# Effects of Marine Natural Products on Liver Diseases

**DOI:** 10.3390/md22070288

**Published:** 2024-06-21

**Authors:** Yandi Sun, Yansong Dong, Xiaohang Cui, Xiaohe Guo, Juan Zhang, Chong Yu, Man Zhang, Haifeng Wang

**Affiliations:** 1Department of Traditional Chinese Materia Medica, Shenyang Pharmaceutical University, Shenyang 110016, China; 15842747934@163.com (Y.S.); dongyansong2023@163.com (Y.D.); cuixiaohang1860@163.com (X.C.); xiao154053@163.com (X.G.); juan159521@163.com (J.Z.); yuchong0408@163.com (C.Y.); 2School of Functional Food and Wine, Shenyang Pharmaceutical University, Shenyang 110016, China; 15094256198@163.com; 3Guangxi Key Laboratory of Marine Natural Products and Combinatorial Biosynthesis Chemistry, Guangxi Academy of Sciences, Nanning 530007, China

**Keywords:** liver disease, marine natural products, ALD, NAFLD, mechanism

## Abstract

The prevention and treatment of liver disease, a class of disease that seriously threatens human health, has always been a hot topic of medical research. In recent years, with the in-depth exploration of marine resources, marine natural products have shown great potential and value in the field of liver disease treatment. Compounds extracted and isolated from marine natural products have a variety of biological activities such as significant antiviral properties, showing potential in the management of alcoholic liver disease (ALD) and non-alcoholic fatty liver disease (NAFLD), protection of the liver from fibrosis, protection from liver injury and inhibition of the growth of hepatocellular carcinoma (HCC). This paper summarizes the progress of research on marine natural products for the treatment of liver diseases in the past decade, including the structural types of active substances from different natural products and the mechanisms underlying the modulation of different liver diseases and reviews their future prospects.

## 1. Introduction

The liver, as the core metabolic organ of the human body, is involved in nutrient metabolism, blood volume regulation, immune support and growth signaling control, as well as the breakdown of drugs and toxins, the storage and regulation of glucose, the processing of lipids and cholesterol, and the metabolism of proteins and amino acids [1]. Although the liver is powerful, it also faces many threats. Currently, studies have identified several risk factors that are closely associated with the development of liver cancer, and viral infection is one of the primary factors for the development of liver cancer. In particular, chronic hepatitis B virus (HBV) and hepatitis C virus (HCV) infections have been widely recognized as the main causative factors of liver cancer. These viruses promote abnormal proliferation and malignant transformation of hepatocytes through different mechanisms, thus increasing the risk of liver cancer. Poor lifestyle habits, especially long-term heavy alcohol consumption, are also important risk factors for liver cancer. The metabolism of alcohol produces toxic substances, which cause direct damage to the liver and trigger alcoholic liver disease, which may then develop into liver cancer [2]. In addition, metabolic diseases such as obesity and diabetes mellitus are also closely related to the increased risk of liver cancer, and these diseases may lead to insulin resistance, abnormal fat metabolism, etc., which promotes the occurrence of liver cancer (Figure 1). 

Recently, liver diseases have become a global public health challenge [3], with a wide variety of etiologies and manifestations that pose a serious threat to human health. Among them, viral hepatitis is the second most lethal infectious disease, claiming 1.3 million lives annually. Specifically, reports indicate that 254 million people are currently living with hepatitis B, another 50 million are battling hepatitis C, and 1 million new infections occur each year [4]. Viral hepatitis is notable for its ability to cause abnormal liver function, which can lead to severe organ damage [5]. Alcoholic liver disease also carries a high risk and usually progresses to liver failure and cirrhosis [6]. Non-alcoholic liver disease is characterized by the development of fatty liver, inflammation and fibrosis, and it is also associated with an increased risk of cardiovascular disease and stroke [7]. Liver fibrosis disrupts the normal structure and function of the organ, thereby reducing its metabolic and detoxification capacity. This damage can adversely affect other organ systems and, in severe cases, eventually lead to multi-organ failure [8]. Hepatocellular carcinoma is a particularly devastating form of liver disease that usually leads to serious complications such as bleeding, ascites and cachexia and has a high mortality rate. It has become the third leading cause of cancer-related deaths [9]. Given the enormous health burden and social impact of liver disease, the treatment and prevention of liver disease are particularly critical and important.

The unique metabolism of the liver and its relationship with the gastrointestinal tract make it an important target for drug and exogenous substance toxicity, making the need for more drug or lead compound therapies a current scientific priority.

Although efficient treatments with few side effects are currently available for typical hepatitis viruses, the development of preventive vaccines remains a major challenge in medicine. For alcoholic liver disease, corticosteroids are used, but their poor acceptance and persistent mortality limit their application [10]. Recent studies have focused on the neutralization of cytokines such as hexoketone cocaine and anti-tumor necrosis factor antibodies for the treatment of severe alcoholic hepatitis. Despite the role of antioxidants in treatment [11], there are no universally accepted therapies that address all stages of alcoholic liver disease. There are no approved drug therapies for NAFLD. Although the efficacy of vitamin E and pioglitazone in non-alcoholic steatohepatitis, a progressive form of NAFLD [12], has been demonstrated, the search for more effective treatments is still needed. The search for new drug sources from marine natural products may be an important direction for the future treatment of NAFLD. The therapeutic efficacy of treatment for HCC is limited by resistance to conventional treatment and tumor recurrence after treatment. Although the pathological mechanisms of liver injury and liver fibrosis are complex and the therapeutic targets are different, marine natural products with abundant bioactive substances have been found to have significant anti-inflammatory and antioxidant effects in recent years; these products can alleviate oxidative stress in the liver and promote the repair and regeneration of the liver, so the treatment of liver diseases with marine natural products has become a new research avenue.

The marine environment, as a treasure trove of wide-ranging biological activity, provides a valuable resource for research in the field of liver disease treatment [13]. Given the uniqueness of marine ecological conditions, such as high salinity, high stress, low temperature, low oxygen availability and light limitations [14], marine organisms have developed metabolic and immune systems that are distinct from those of terrestrial organisms over a long evolutionary process [15]. These immune systems have evolved the ability to fight against a wide range of marine pathogens and toxins, providing a wealth of information for the development of antiviral and anticancer drugs [16]. 

Currently, approximately 20 active substances of marine origin are in clinical use for cancer treatment and antibacterial and antiviral applications. In recent years, there has been a proliferation of new compounds in the field of marine natural products, often tested for cancer cell toxicity, antibacterial and anti-inflammatory activity (Figure 2). Alkaloids, polyketides, terpenoids, phenols and polysaccharides, as well as other compounds of novel structure contained in fungi, bacteria, actinomycetes, plants and animals from the vast oceans have a wide range of biological activities, such as significant antiviral properties, which inhibit the replication of the hepatitis C virus and the hepatitis B virus [17], and protect the liver from fibrosis through various mechanisms. In addition, these marine natural products have shown potential in the management of alcoholic liver disease and non-alcoholic fatty liver disease and protection against liver injury due to their anti-inflammatory and antioxidant properties [18]. More notably, they exhibit anticancer activity against hepatocellular carcinoma cells by inducing cell death and inhibiting tumor growth.

## 2. Liver Diseases

### 2.1. HCV and HBV

Hepatitis C virus is a major cause of chronic hepatitis, liver cirrhosis, and hepatocellular carcinoma [19]. The nonstructural protein (NS3) helicase of HCV, which is crucial for viral replication, holds promise as a target for novel antiviral drugs. NS3 plays a central role in HCV replication, encompassing both helicase and protease functions. The N-terminal third of NS3 has serine protease activity and processes the C-terminal portion of the protein, which contains nonstructural proteins. The helicase function of the remaining part of NS3 involves ATPase and RNA-binding activities, which are essential for unwinding double-stranded RNA during viral genomic RNA replication [20]. 

Marine sponges have shown impressive potential in the field of anti-hepatitis C virus research in recent years. Important progress has been made in the study of the marine sponge, *Spongia irregularis*. Researchers found that the extract of its ethyl acetate fraction exhibited a significant inhibitory effect on HCV, with an IC_50_ value of 12.6 μg/mL. Further chemical and metabolomic analyses revealed several potentially active compounds, among which nakijiquinone F (**1**) showed the best binding affinity for molecular docking with the NS3 deconjugating enzyme, showing great promise as an anti-HCV drug [21]. In addition, the study by Karimov et al. focused on *Aspergillus versicolor*, a fungus living in Red Sea sponges, and they found that the ethyl acetate extract of this fungus was effective in inhibiting the activity of the HCV NS3/4A protease. Among them, two diketopiperazines, cyclic (L-proline-L-enine) (**2**) and cyclic (L-tyrosine-L-proline) (**3**), exhibited strong inhibitory effects with IC_50_ values of 13.4 and 8.2 μg/mL, respectively [22]. Four new acyclic sesquiterpenes conjugated with sugar alcohols, acremosides A (**4**) and C–E (**5**–**7**), were isolated from cultures of the sponge-related fungus *Acremonium* sp. IMB18-086 grown in the presence of heat-killed *Pseudomonas aeruginosa*; these compounds were highly inhibitory against hepatitis C virus, with EC_50_ values ranging from 4.8 to 8.8 μM [23]. 

Researchers successfully isolated two novel alkaloids, designated raistrickindole A (**8**) and raistrickin (**9**), from the marine fungus *Penicillium raistrickii* IMB17-034. These compounds exhibited inhibitory activity against hepatitis C virus, with EC_50_ values of 5.7 and 7.0 μM, respectively [24]. Additionally, a novel pyrazine derivative, trypilepyrazinol (**10**), was isolated and characterized from the marine fungus *Penicillium* sp. IMB 17-046, which also showed antiviral activity against hepatitis C virus with an IC_50_ value of 7.7 μM [25]. These studies suggest that marine fungi are a rich source of bioactive metabolites with anti-HCV viral properties.

Hepatitis B is a highly prevalent and serious chronic disease caused primarily by the hepatitis B virus [26]. This virus is a major precursor to cirrhosis and HCC [27], both of which typically result in considerable morbidity and mortality. Hepatitis B surface antigen (HBsAg) plays a key role in the pathogenesis of this disease [28]. This antigen is a viral capsid protein that is present on the outside of HBV particles. The presence of HBsAg is a reliable indicator of HBV infection and helps in early detection and subsequent intervention as a key recognition marker of the virus [29]. cccDNA is essential for the persistence and replication of HBV in hepatocytes. It exists in the form of microchromosomes in the nucleus of hepatocytes and serves as a template for viral transcription. In the presence of cccDNA, HBV can continuously synthesize new viral RNA and proteins, which are then assembled into new viral particles that are released from hepatocytes and continue to infect other hepatocytes (Figure 3) [30].

Junceellolide B (**11**), a marine natural product from an in-house library, was used as an HBV inhibitor. Li et al. reported that junceellolide B reduced HBsAg and HBeAg production in HBV-infected cells, and HBV DNA, RNA, and HBeAg secretion in HepAD38 cells. It inhibited HBV RNA transcription and reduced the number of cccDNA-transcribed products, downregulating RNA polymerase II-related transcription factors. Junceellolide B is a promising transcription inhibitor of cccDNA for anti-HBV agent development [31].

Li et al. isolated a novel anthraquinone derivative, (−)-2′*R*-1-hydroxyisorhodoptilometrin (**12**), from the acidic fermentation broth of *Penicillium* sp. OUCMDZ-4736, a fungus isolated from the root sediment of mangrove plants, which showed more potent activity against hepatitis B virus than the positive control drug, and effectively inhibited the secretion of HBsAg and HBeAg from HepG2.2.15 cells with an IC_50_ value of 4.63 μM [32]. 

Narula et al. evaluated the potential of the marine anti-microbial cell-penetrating peptide Tachyplesin (Tpl) as an anti-HBV drug. Tpl significantly reduced the expression of HBV proteins, including HBsAg and HBeAg. In addition, Tpl reduced the levels of HBV prenuclear RNA and HBV pregenomic RNA, suggesting that the inhibitory effect of Tpl occurs during the early stages of HBV replication, including viral transcription. In addition, Tpl significantly reduced hepatitis B virus secretion. In conclusion, this finding emphasizes its potent anti-HBV activity at non-cytotoxic concentrations [33]. 

Metachromin A (**13**), a merosesquiterpene isolated from the marine sponge *Dactylospongia metachromia*, was found to effectively inhibit HBV production and viral promoter activities. It suppressed HBV production with an EC_50_ value of 0.8 µM. Metachromin A also significantly reduced the amount of HNF4*α* protein, which may have contributed to its antiviral activity. This compound shows promise as a potential antiviral agent for HBV-related liver disorders [34]. 

Yamashita et al. extracted two polybrominated diphenyl ethers (PBDEs) with strong inhibitory properties and low cytotoxicity against HBV from the marine sponge *Dysidea* sp. 3,5-dibromo-2-(2,4-dibromophenoxy)-phenol (**14**) and 3,4,5-tribromo-2-(2,4-dibromophenoxy)-phenol (**15**) inhibited HBV core promoter activity and HBV production in a dose-dependent manner with EC_50_ values of 0.23 and 0.80 µM, respectively. These two PBDEs proved to be potential candidates for the development of anti-HBV drugs [35]. The chemical structures of compounds **1**–**15** are shown in Figure 4. Many of the above compounds have small IC_50_ or EC_50_ values, and these marine natural products provide strong candidates for the future development of novel and potent antiviral drugs (Table 1).

### 2.2. ALD

Alcoholic liver disease is a liver disease caused by prolonged and heavy drinking, the incidence of which is on the rise worldwide and has become a serious public health problem [36]. ALD has a complex pathology involving multiple aspects of alcohol metabolism, oxidative stress, and inflammatory response, which ultimately leads to hepatocellular injury, steatosis, fibrosis, and even hepatocellular carcinoma. Adenosine monophosphate-activated protein kinase (AMPK), a key energy sensor, can lead to the downregulation of the adipogenic gene sterol regulatory element-binding protein 1 (SREBP-1) and the upregulation of the fatty acid oxidation gene peroxisome proliferator-activated receptor-alpha (PPAR*α*) by affecting key transcription factors. Therefore, the strategic localization of AMPK represents a promising therapeutic approach, and thus, a deeper understanding of the pathogenesis, clinical manifestations, and therapeutic approaches of ALD is important for the prevention and treatment of ALD [37].

Peptide components found in the ocean can alleviate ALD through antioxidant activity and reducing oxidative stress. Oyster meat contains a large number of functional peptides, glycogen, and polyunsaturated fatty acids, and a variety of bioactivities including antioxidant [38] and anti-inflammatory activities [39] have been obtained from oyster meat extracts. Oyster peptide (OP) significantly improved biochemical indices of ALD in mice, such as serum glutamyl transferase (AST), alanine aminotransferase (ALT), *γ*-glutamyl transferase (GGT), reactive oxygen species (ROS), malondialdehyde (MDA), and triglyceride (TG) activities, superoxide dismutase (SOD) activity and glutathione (GSH) concentration; decreased oxidative stress and inflammation; and increased the activity of antioxidant enzymes. Mechanistic studies showed that OP exerted protective effects by up-regulating the expression of the antioxidant-related gene Nrf2 and down-regulating the mRNA expression of the inflammation-related genes NF-*κ*B, TNF-*α*, and IL-6. This suggests that oyster peptides may be natural and effective hepatoprotective agents and are expected to be applied in the treatment of ALD (Figure 5) [40].

In addition, Seeker’s muscle clam also possesses antioxidant and alcohol dehydrogenase stabilizing activities that attenuate alcoholic liver injury, enhance hepatic alcohol dehydrogenase (ADH) activity and the antioxidant defense system, and reduce oxidative stress in mice. Peptidomics analysis revealed a variety of favorable peptides, some of which were synthetic peptides with in vitro bioactivity. Thus, Seeker’s muscle clam is a potential source of bioactive peptides for the prevention of alcoholic liver injury [41].

Marine polysaccharide extracts increase antioxidant levels to reduce the amount of oxidized substances produced during alcohol metabolism, protecting liver cells from further damage. Algal oligosaccharides (AOSs) extracted from seaweed can increase the antioxidant levels of GSH and SOD and ameliorate the elevated production of MDA. AOS can also increase hepatic ADH activity, which provides a new strategy for reducing alcohol-induced hepatotoxicity [42].

*Enteromorpha prolifera* polysaccharide (PEP), a seaweed polysaccharide extracted from the green alga seaweed *Enteromorpha prolifera*, possesses a variety of biological activities, including antioxidant activity, due to its sulfated heteropolysaccharide properties. In an alcohol-induced liver injury model in C57BL/6 mice, PEP significantly reduced the serum ALT and AST levels, improved liver histology, and reduced microvesicular steatosis and ballooning. EP also enhances alcohol metabolism by regulating metabolite-related enzymes (ADH and ALDH), decreases CYP2E1 expression, and enhances hepatic antioxidant capacity by regulating the Nrf2/HO-1 signaling pathway, demonstrating the protective effect of EP against alcohol-induced liver injury [43].

Laminarin extracted from brown algae also has ameliorative effects on ALD. Daily administration of 100 mg/kg Laminarin to mice significantly attenuated alcohol-induced liver injury and improved liver function. The mechanism may involve the regulation of biological pathways such as the antioxidant, WNT and cAMP pathways, which modulate the expression of related genes. This provides a new idea for the prevention of alcoholic liver disease, and is of positive significance for the development of functional foods [44].

Sulfated sea cucumber polysaccharides (SCSPs) extracted from Japanese ginseng were found to protect the liver. To improve the solubility, photocatalytic reaction depolymerization was used to obtain low molecular weight products (dSCSPs). In vivo experiments showed that the addition of either SCSP or dSCSP to Baijiu could alleviate alcoholic liver injury in mice, and dSCSP was more effective at reducing liver MDA levels. Therefore, the addition of sea cucumber polysaccharides or their low-molecular-weight derivatives to white wine helps to reduce alcoholic liver injury [45].

*Melosiraeus nummuloides* is a microalga belonging to the diatom genus Diatoms in the Melosiraceae family. *M. nummuloides* ethanol extract (MNE) results in a nutritionally favorable fatty acid profile characterized by a greater ratio of omega-3 to omega-6 fatty acids than does the original raw material. By using alcohol-treated HepG2 cells and chronic binge alcohol-fed mice, it was found that MNE significantly reduced AST and ALT levels and decreased PPAR*α* in chronic alcoholics, in addition to markedly inhibiting the mRNA expression of the inflammatory cytokines TNF-*α*, IL-1*β* and IL-6. Therefore, MNE has a good ability to alleviate hepatic steatosis, oxidative stress and inflammation [46].

In recent years, marine terpenoids have been shown to ameliorate alcohol-induced liver injury by increasing the levels of associated serum aminotransferases and pathways.

Aplysin (**16**) is a marine bromo sesquiterpene extracted from the red algae *Laurencia tristicha*. It improved liver tissue structure and biochemical indices and attenuated DNA damage. Mechanistically, aplysin attenuates oxidative stress and regulates the expression of Bcl-2 and Bax, genes associated with apoptosis, providing significant protection against alcohol-induced liver injury [47].

Fucoxanthin (Fx) (**17**) is a seaweed extract that has been shown to reduce AST and ALT activity, decrease serum and liver daily oil triglycerides, activate AMPK and subsequently regulate the KEAP1/Nrf2/ARE signaling pathway to exert antioxidant effects and stimulate the PGC1*α*/NRF1 axis to promote mitochondrial biogenesis. These combined effects contribute to the amelioration of hepatic steatosis caused by metabolic disorders by Fx. These findings provide valuable insights into the use of Fx as a therapeutic strategy for the treatment of ALD [48], and it is interesting to note that Fx can also play an important role in NAFLD; therefore, Fx is considered a valuable natural product for treating liver disease [49].

Marine phenolic components clear free radicals and reduce oxidative stress. 7-Phloro-eckol (7PE) (**18**), a polyphenol isolated from the seaweed *Ecklonia cava*, was shown to attenuate alcohol-induced HepG2 cytotoxicity, DNA damage, and oxidative stress by modulating the expression of various related proteins, such as glutathione (GSH), superoxide dismutase (SOD), and B-cell lymphoma 2 (Bcl-2). damage and oxidative stress [50].

### 2.3. NAFLD

Non-alcoholic fatty liver disease is an emerging global public health threat affecting approximately 1.8 billion people worldwide with a prevalence of 20–30% [51], mainly because of its strong association with a preference for high-fat foods and metabolic syndrome, coupled with the lack of effective pharmacologic treatment [52]. Its pathogenesis involves multiple factors, including insulin resistance, abnormalities in fat metabolism, oxidative stress and genetic susceptibility.

Abnormal lipid metabolism leading to excessive fat deposition in the liver is the core feature of NAFLD, so regulating lipid metabolism has an important role in treating the development of NAFLD (Figure 5). Extracts of marine algal natural products can effectively inhibit the expression of fatty acid synthetase and cholesterol biosynthesis-related transcription factors, and regulating phospholipid metabolism has an important role in the development of NAFLD.

Susabinori lipid (SNL) enriched with eicosapentaenoic acid (EPA), a polar lipid extracted from the marine red alga Susabinori (*Pyropia yezoensis*), had hepatoprotective effects on a mouse model of type 2 diabetes. SNL upregulated 15 genes associated with SNL affected arachidonic acid (AA) and linoleic acid (LA) metabolism and increased EPA accumulation, which contributed to the amelioration of hepatic steatosis. These findings provide molecular mechanistic insights into the beneficial effects of SNL [53].

GCSW210, a subcritical aqueous extract of the red alga *Gracilaria chorda* (GC), also significantly reduced lipid accumulation in oleic acid-induced hepatic steatosis cells and in high-fat diet-induced obese mice. GCSW210 improves glucose metabolism through the modulation of adipogenic factors and the cholesterol biosynthesis-associated transcription factors, SREBP-2 and LDL receptor and improves insulin resistance. In addition, GCSW210 enhances the phosphorylation of AMPK and improves insulin signaling, and the active substance 5-hydroxymethylfurfural may be a key component of GCSW210. Therefore, GCSW210 has the potential to prevent and treat metabolic syndrome diseases [54].

Krill oil (KO) extracted from Antarctic krill (*Euphausia superba*) is a rich alternative source of *n*-3 PUFAs [55]; it contains a high percentage (30–65%) of docosahexaenoic acid (DHA) and EPA in the form of phospholipids (mainly phosphatidylcholine) and has demonstrated therapeutic potential for the treatment of obesity and the metabolic syndrome in a mouse model of a high-fat diet [56]. KO significantly reduced body weight, decreased adipocyte hypertrophy, and decreased serum transaminase and cholesterol levels. Mechanistically, KO inhibits lipid synthesis and increases fatty acid β-oxidation by regulating gene expression and the CB1/lipocalin/ceramide pathway, thereby more effectively inhibiting hepatic TAG accumulation [57]. Therefore, krill oil has demonstrated superior results in the treatment of NAFLD [58]. 

Marine phenolics can also improve liver function by regulating various factors such as lipid metabolism. The screening of a library of marine natural compounds revealed that the phenolic AMPK activator CHNQD-0803 was isolated from a fungus (*A. candius*). It can directly bind to AMPK and activate its activity and was found to inhibit lipogenesis, reduce inflammation, and ameliorate liver injury and fibrosis in a variety of cellular and animal models. Therefore, CHNQD-0803 is also a novel potential candidate for non-alcoholic steatohepatitis (NASH) treatment [59]. *Posidonia oceanica* is a marine plant, and UPLC characterization analysis revealed that POE consists of 88% phenolic compounds. It has cell-safe bioactivities, including the ability to trigger autophagy. POE counteracts NAFLD by modulating the expression of autophagy markers, stimulating autophagic flux, and reducing lipid accumulation in HepG2 cells. Therefore, *P. oceanica* may be a promising natural therapy for the management of NAFLD [60].

Xyloketal B (Xyl-B) (**19**) is a unique ketal compound isolated from the mangrove fungus *Xylaria* sp. in South China that attenuates NAFLD-associated lipid accumulation. Through quantitative proteomics and experimental methods, Xyl-B was found to have a protective effect on fatty liver in a high-fat diet mouse model. Among the keys to Xyl-B’s ability to ameliorate hepatic steatosis in mice was the up-regulation of PPAR*α* downstream enzymes associated with fatty acid oxidation. In addition, Xyl-B also exhibited anti-inflammatory and anti-fibrotic effects in hepatitis and liver fibrosis models. These results suggest that Xyl-B ameliorates different stages of NAFLD by activating the PPAR*α*/PGC1*α* signaling pathway [61].

Furazone (**20**), isolated from the marine fungus *Setosphaeria* sp. SCSIO41009, was found to reduce lipid accumulation by targeting LXR*α* and PPAR*α*, two key regulators of lipid metabolism, in in vitro studies due to its unique bioactivity, and thus furazone has the potential to be a potent drug candidate for the treatment of NAFLD. These findings provide new ideas for the treatment of NAFLD [62]. 

The fumonisin derivatives 12R,13S-dihydroxyfumitremorgin C (**21**), and tryprostatin A (**22**) isolated from the sponge fungus *Aspergillus* sp. SCSIO 41420 may have unique chemical structures and biological activities that allow for more effective activation of LXR*α*, as well as lowering cholesterol levels and improving lipid metabolism. However, it is important to note that while the activation of LXR*α* may be beneficial in NAFLD, over-activation may also lead to side effects such as hepatic steatosis and insulin resistance. Therefore, when developing drugs targeting LXR*α*, their therapeutic effects need to be carefully balanced against the potential risks [63].

Marine-activated protein peptides have a variety of physiological functions such as antioxidant, anti-inflammatory and immune modulatory effects. These functions may have positive implications for improving the condition of NAFLD patients. *Arthrospira platensis* is commonly known as Spirulina, and most Spirulina has been shown to be associated with c-protein (C-PC) activity, which is protective against high-fat diet-induced NASH and is closely related to its role in reducing inflammation and oxidative stress, maintaining glucose tolerance, and ameliorating hepatotoxic bile acids. Supplementation of *Arthrospira platensis* phycobiliprotein peptide extracts (PPEs) to rats with high-fat diet-induced NAFLD reduced body weight, decreased hepatic lipid droplet accumulation, and modulated lipid metabolism and intestinal metabolites to improve NAFLD, demonstrating significant therapeutic effects [64].

Marine polysaccharide components are involved in the immunomodulatory effects of the organism, and chitosan, a collagen-based material derived from marine biocomposites [65], is a cationic alkaline oligosaccharide [66], which possesses multiple physiological functions, including antimicrobial, anti-obesity, and hypolipidemic effects. It has been shown to significantly improve body weight, fat levels and hepatic steatosis in obese animals [67]. By addressing the key “liver-gut axis” disorder in the pathogenesis of NAFLD, we demonstrated that chito-oligosaccharides with molecular weights ≤ 3000 were able to alleviate hepatic steatosis by regulating the intestinal flora and improving the hepatic LPS/TLR4/NF-*κ*B inflammatory pathway. This finding provides strong support for the use of chitosan in health applications. The consumption of U.S.-grown sugar kelp effectively reduced obesity, metabolic disorders, hepatic steatosis, inflammation, and fibrosis while increasing energy expenditure and decreasing the hepatic expression of the macrophage marker adhesion G-protein-coupled receptor E1 (Adgre1) and the macrophage M1 marker integrin x (Itgax) in a NASH mouse model [68].

Carotenoids are a class of natural pigments widely found in plants and marine organisms, and they have strong antioxidant capacity. *Nitzschia laevis* extract (NLE), a fucoidan-rich microalgal extract whose main components are carotenoids, has hypolipidemic effects on a high-fat diet-induced steatosis mouse model and palmitic acid-treated HepG2 cells. The experimental results showed that NLE significantly reduced body weight and hepatic steatosis and inhibited cellular lipid accumulation in mice, and NLE also achieved lipid-lowering effects by enhancing hepatic mitochondrial functions, such as increasing the oxygen consumption rate and mitochondrial membrane potential, as well as inhibiting fatty acid synthesis. Therefore, it is an extremely promising natural product for the prevention of NAFLD [69]. The chemical structures of compounds **16**–**22** are shown in Figure 6.

### 2.4. Liver Fibrosis

Liver fibrosis is a key factor in the worsening of chronic liver diseases such as viral hepatitis and fatty liver [70]. Without effective intervention, 75–80% of patients can develop cirrhosis, which seriously jeopardizes human health [71]. The extracellular matrix (ECM) synthesis and the transformation of myofibroblasts (MFs) induced by the activation of hepatic stellate cells (HSCs) are key factors in liver fibrosis. The reversibility of these factors also provides an important research target for the reversal of liver fibrosis. The mechanism of liver fibrosis is complex and involves histopathology, cytology, cytokines and their regulation at the molecular level. In the study of anti-fibrotic mechanisms, a variety of natural marine extracts have demonstrated remarkable efficacy.

Transforming growth factor *β*1 (TGF-*β*1) and the Smad pathway are key to preventing liver fibrosis [72], and marine polysaccharide natural products can effectively inhibit transforming growth factor and hepatic stellate cell activation and extracellular matrix deposition to combat liver fibrosis.

Xu et al. isolated propylene glycol sodium sulfate (PSS), a sulfated polysaccharide derivative, from a natural extract of brown algae, which exhibited significant antifibrotic effects in fibrotic mice and cellular models by inhibiting the TGF-*β*1/Smad pathway and facilitating ECM catabolism. It also exerts anti-autophagic effects by inhibiting the JAK2/STAT3 pathway to further counteract the process of liver fibrosis and liver injury [73].

Additionally, in a carbon tetrachloride-induced liver injury model in rats, another fucoidan (FE) extracted from Sargassum is a sulfated polysaccharide that can reduce hepatic enzyme activity, inflammatory cell infiltration, and collagen fiber deposition, and modulate the gene expression of cytokines such as interleukin *β*1 (IL-*β*1), TNF-*α*, TGF-*β*1, Smad-3, Smad-2, collagen 1*α*1 (col1*α*1). Thus, FE can also exert antifibrotic and anti-inflammatory effects by inhibiting the TGF-*β*1/Smad pathway [74]. Therefore, fucoidan has become a promising natural marine product for the treatment of liver fibrosis.

Cholesterol accumulation is also a risk factor for hepatic fibrosis. Gm-SGPP with pentasaccharide structure extracted from Atlantic cod Gadus morhua has the potential to inhibit hepatic fibrosis. The hepatic fibrosis model mice were injected intraperitoneally with 2.5% CCl_4_ (10 mL/kg) and orally administered Gm-SGPP (500 mg/kg) for 30 days. The results showed that Gm-SGPP attenuated oxidative liver injury and lipid metabolism disorders, and reduced hepatocellular necrosis and lipid droplet accumulation. Gm-SGPP increased the amount of bile acids and altered the composition of bile acids by increasing the expression of cholesterol 7a-hydroxylase (CYP7A1) and sterol 27-hydroxylase (CYP27A1), which inhibited the expression of the ileoadrenal lipid X receptor (FXR), and accelerated cholesterol conversion. Moreover, Gm-SGPP prevents hepatic fibrosis by down-regulating the TLR4-mediated TGF-*β*/Smad pathway, and to the best of our knowledge, this is the first report showing that Gm-SGPP prevents hepatic fibrosis by reducing cholesterol accumulation [75].

In recent years, novel heteroterpenoid derivatives derived from marine fungi have been shown to significantly inhibit the expression of liver fibrosis-related genes caused by certain predisposing factors. These findings suggest that these marine terpenoid components have anti-hepatic fibrosis potential and can be used to prepare anti-hepatic fibrosis drugs.

The protective effects of an organic extract of Red Sea soft coral (*Sarcophyton glaucum*) and its major steroidal constituents against acetaminophen-induced liver fibrosis in rats were evaluated. The effects were assessed by detecting liver function parameters, tumor markers and gene expression in liver tissues, as well as by observing pathological changes in liver tissues. Moreover, the interaction of marine steroids with the detoxification enzymes glutathione S-transferase (GST) and SOD were investigated computationally, and two promising steroid modulators were identified, dissesterol (**23**) and glaucasterol (**24**), that have good binding ability to GST and SOD, in addition to the double bond on the B ring of the tetracyclic steroid system in the structure and the cyclopropane ring on the side chain, which seem to be the key structural elements for the enhancement of hepatic protective activity of the tetracyclic steroid system [76].

Aspermeroterpene B (**25**) isolated from the marine-derived fungus *Aspergillus terreus* GZU-31-1 effectively inhibited hepatic stellate cell activation and collagen deposition, induced apoptosis, and activated the antioxidant enzymes heme oxygenase 1 (HO-1) and NAD(P)H quinone dehydrogenase 1 (NQO-1). In in vitro experiments, the compound significantly inhibited hepatic stellate cell activation by targeting the Nrf2 signaling pathway at a concentration of 5 μM, demonstrating a potential anti-hepatic fibrosis effect [77].

Marine alkaloid constituents are also hepatoprotective, and ovalbumin A (**26**), which is isolated from sea urchin eggs, markedly reduces the levels of collagen fibers and fibrosis markers, such as transforming growth factor (TGF-*β*), *α*-smooth muscle actin (*α*-SMA), and TIMP-1, and maintains intracellular redox reactions by modulating GGT expression homeostasis. These findings suggest that ovalbumin A has potential as a new drug or dietary supplement for the treatment of liver fibrosis [78].

Other marine extracts have been found to have inhibitory effects on liver fibrosis. The protective effects of two marine macroalgae (*C. racemosa* and *P. pavonia*) against carbon tetrachloride-induced liver fibrosis in rats were investigated. By continuously injecting rats with carbon tetrachloride and concomitantly administering aqueous extracts of seaweeds, these extracts were found to significantly reduce the serum transaminase, alkaline phosphatase and bilirubin levels, improve the oxidative stress, stabilize liver cell morphology, and improve renal function and lipid metabolism. The results suggest that these seaweed extracts have a certain inhibitory effect on liver fibrosis [79].

Astaxanthin (3,3′-dihydroxy-*β*,*β*′-carotene- 4,4′-dione, ASX) 3S-3′S Astaxanthin from marine organisms is a carotenoid derived mainly from oxygenated non-vitamin a sources [80] and exists mainly in the form of 3S, 3S, or 3R, 3′R. 3S-3′S Astaxanthin (**27**), 3R-3′S Astaxanthin (**28**) and 3R-3′R Astaxanthin (**29**) have been used in vivo for the prevention and treatment of several systemic diseases. Researchers have demonstrated that astaxanthin is a multifaceted antifibrotic compound that protects the liver from fibrotic damage by inhibiting the activation of resting hematopoietic stem cells and decreasing the production of reactive oxygen species, as well as by increasing the expression of the related factor NrF2 [81]. It also restores the activity of damaged antioxidant enzymes [82] and inhibits the formation of extracellular matrix through the TGF-*β*1/Smad3 pathway [83], whereas apoptosis and autophagy can be directly related to the survival status of hepatic stellate cells. In addition, astaxanthin down-regulates the expression of histone deacetylase, which further inhibits the occurrence of hepatic fibrosis [84]. The hydroxyl group present in the end-loop structure can form an ester with fatty acids, thus enhancing the antioxidant activity of the natural extract of astaxanthin.

### 2.5. Liver Injury

Liver injury is frequently caused by various factors, including viral hepatitis, ischemia reperfusion injury (IRI) of the liver, and drug-induced liver damage (DILI). Additionally, it is crucial to consider liver damage resulting from prevalent chronic liver conditions such as liver fibrosis and alcoholic/non-alcoholic fatty liver disease [3]. 

Fucoidan, a sulfated polysaccharide complex derived from marine brown algae [85], plays a key role in restoring the activity of cells with excessive ROS production, such as hepatocytes, due to its potent antioxidant capacity [86]. In response to ROS production triggered by acrolein, a toxic metabolite of cyclophosphamide (CTX), and liver injury in vivo due to oxidative stress, fucoidan activated Nrf2/HO-1 signaling, an important mediator of cellular responses during oxidative stress. Activated Nrf2 downstream proteins are involved in cellular antioxidant mechanisms (e.g., HO-1, GSH, GLCM, etc.). Therefore, fucoidan may ameliorate CTX-induced hepatic and renal injuries by upregulating the Nrf2/HO-1 pathway and inhibiting the TLR4/NF-*κ*B pathway [87]. Moreover, in a study of thioacetamide (TAA)-induced liver injury in male C57BL/6 mice, mice treated with fucoidan showed improvements in body weight, food intake, and hepatic antioxidant enzymes, as well as a decreasing trend in serum indicators of liver injury such as ALT, AST, TNF-*α*, IL-1*β*, and CRP. The mRNA expression of inflammation-related genes such as COX-2 and iNOS was also reduced. In addition, fucoidan effectively inhibits the synthesis of type 1 collagen and *α*-SMA, which tend to be up-regulated in damaged hepatocytes, and thus promote the transformation of HSCs into myofibroblasts, leading to liver fibrosis [88]. In conclusion fucoidan protects against liver injury by inhibiting inflammatory responses and oxidative stress [89].

The marine extract *Galaxaura oblongata* reduces inflammatory responses and oxidative stress by lowering the levels of serum cytokines such as NF-*κ*B, MPO, and LPO. In addition, it ameliorated hepatocyte apoptosis by inhibiting protein tyrosine kinase signaling, and was protective in mice with acute liver injury [90]. Similarly, LM49 (**30**), an active marine halogenated phenol extracted from marine algae, gastropods and sponges, protected mice against LPS-induced acute liver injury. This protective effect was attributed to its anti-inflammatory signaling pathway and subsequent induction of M2-type Kupffer cells. These findings suggest that both *G. oblongata* extract and LM49 have potential efficacy in the treatment of LPS-induced acute liver injury [91].

HN-001 (**31**), a natural substance extracted from the marine fungus *Aspergillus* sp. C1, has hepatoprotective effects on mice after long-term administration. Its ability to alleviate liver injury was associated with the inhibition of the PLA2/IRE-1*α*/XBP-1s axis and c-Jun N-terminal kinase (JNK) signaling. Specifically, HN-001 reversed palmitic acid (PA)-induced hepatocyte death in a dose- and time-dependent manner. Toxic lipids can injure hepatocytes through the endoplasmic reticulum (ER) stress response, during which IRE-1a activates proteins such as spliced XBP-1 (XBP-1s) and JNKCDK2, which have been implicated in the death of both parenchymal and nonparenchymal cells of the liver. The ability of HN-001 to regulate these processes suggests that it attenuates hepatic steatosis, hepatic inflammation, liver fatigue, degeneration, liver injury, inflammation, and liver fibrosis [92]. 

Hepatic ischemia/reperfusion injury is a pathologic inflammatory response that occurs after a period of hepatic ischemia when hepatic blood flow is restored, and can lead to severe hepatic dysfunction, multi-organ failure, and a systemic inflammatory response syndrome. In the ongoing search for biologically active fungal metabolites, researchers have successfully isolated 1 indole alkaloid, notoamide Q (**32**), from the endophytic fungus *Aspergillus amoenus* TJ507, which exerted a significant protective effect against hepatic injury and apoptosis in a mouse model of hepatic ischemia-reperfusion injury. To quantitatively assess the extent of liver injury, the researchers measured serum levels of ALT, AST, and lactate dehydrogenase (LDH), and pretreatment with notoamide Q significantly reduced serum levels of these enzymes when compared to the untreated IRI group. This finding suggests that notoamide Q has the potential to be a lead compound in the development of therapeutic drugs for liver IRI [93]. The chemical structures of compounds **23–32** are shown in Figure 7. The different effects of compounds on ALD, NAFLD, liver fibrosis and liver injury have been summarized in Table 2.

### 2.6. HCC

HCC is a particularly aggressive form of liver cancer with alarmingly high morbidity and mortality rates [91], and HCC is primarily associated with chronic infections caused by HBV and HCV, as well as non-alcoholic NASH [94]. Many scientists are exploring marine natural products as potential sources of novel therapeutics. 

SOX9 is a transcription factor that is significantly expressed in HCC cancer stem cells and plays a crucial role in promoting cell proliferation and self-renewal [95]. Its oncogenic properties have been demonstrated in a variety of malignancies including HCC [96], and cell cycle protein-dependent kinase 2 (CDK2) [97] interacts with SOX9 to promote cell cycle progression [98]. The marine sponge alkaloid aaptamine (**33**) has been shown to be effective against HepG2 liver cancer cells in various in vitro studies, and 75 µg/mL aaptamine was shown to have a very effective cytotoxic effect on HepG2 cells. Thus, it is a promising drug candidate for the treatment of HCC [99].

The Wnt/*β*-catenin signaling pathway is crucial for the development of hepatoblastoma, a significant neoplastic condition. Among the potential therapeutic agents targeting this pathway, demethylincisterol A3 (Sdy-1) (**34**), a highly degraded sterol, stands out. Derived from the endophytic *Pestalotiopsis* sp. HQD-6 residing in the Chinese mangrove forest Rhizophora mucronata, Sdy-1 exhibits remarkable inhibitory effects on the Wnt/*β*-catenin signaling pathway. Notably, it effectively inhibited the proliferation and migration of human HepG2, while inducing apoptosis and arresting the cell cycle at the G1 phase. These findings indicate that Sdy-1 possesses potent inhibitory effects on the Wnt signaling pathway, making it a promising candidate for antitumor drug development [100].

Oxidative stress is a key factor in the pathogenesis of liver tissue injury and HCC. Chronic tissue injury is usually associated with the production of ROS, and the accumulation of ROS in the liver leads to cellular dysfunction and the emergence of cancer cells [101]. Research has shown that the anticancer effect of heteronemin (**35**) on HCC is associated with ROS-associated MAPK activation and that heteronemin induces HCC death by inhibiting the proliferation of the HCC cell lines HA22T and HA59T via the caspase pathway [102].

Matrix metalloproteinases (MMPs) are a group of zinc-dependent enzymes that are essential for the degradation of ECM, thereby promoting tumor invasion and metastasis. MMP-2 and MMP-9, also known as gelatinases, are the most studied MMPs. The overexpression of MMP-2 and MMP-9 has been associated with advanced tumor-node-metastasis (TNM) staging because they play a role in promoting metastasis, invasion and poor differentiation of tumor cells [103]. Stellettin B (**36**) is an active compound isolated from *Stelletta* sponges. The inhibitory effects of stellettin B on cell migration and invasion in human hepatoma cells are associated with downregulation of the activity and protein expressions of MMP-2 and MMP-9 [104]. The polyphenol-rich leaf extract of *P. oceanica* (POE) prevents intracellular lipid accumulation and blocks the MAPKs/NF-*κ*B axis. The NF-*κ*B pathway is involved in cancer metastasis via regulation of metalloproteinases MMP-2/9, which plays a key role in HCC invasion and metastasis, and consequently reduces MMP-2/9 in HepG2 cells under high glucose (HG) conditions; these cells are used as an in vitro model of HCC [105]. 

Since the HepG2 cell line is derived from the liver tissue of patients with hepatocellular carcinoma, it has become an important model for in vitro studies of HCC [106]. Morphologically and functionally, the HepG2 cell line is similar to human liver tissue, and thus is commonly used to mimic the hepatocellular carcinoma environment in humans. In addition, since the p53 oncogene in HepG2 cells is not mutated, it also provides an important tool for studying the close relationship between p53 and hepatocellular carcinoma, as well as the pathogenesis, diagnosis, and prevention of hepatocellular carcinoma [107]. In addition, (−)-agelasidine A (**37**), a sesquiterpene guanidine compound extracted from the methanolic extract of the sponge *Agelas nakamurai* by Lu et al., has been shown to significantly reduce Hep3B and HepG2 cell viability. Moreover, it triggered endogenous and exogenous apoptosis by inducing endoplasmic reticulum stress. Thus, (−)-agelasidine A is expected to be a potential drug candidate for the treatment of HCC [108]. A novel squamous alkane derivative named phyllospongiane C (**38**) was isolated from the marine sponge *Phyllospongia foliascens*, which all possessed an unprecedented 6/6/6/5 tetracyclic dioquamous scaffold. Notably, it exhibited significant cytotoxic activity against HepG2 cancer cell lines with an IC_50_ value of 9.6 μM [109].

In recent studies, marine fungi and bacteria have emerged as rich sources of bioactive compounds with potential therapeutic applications. Among these, *Chloridium* sp. NBU3282, a marine fungus, was found to produce a unique indole sesquiterpene named asepterpenedol A (**39**). This compound demonstrated significant cytotoxicity against HepG2 cells, suggesting at its potential as a novel scaffold for anti-liver cancer drug development [110]. In addition, the marine-derived *Streptomyces* sp. strain CA-271078 produced a novel near-tearomycin D1 (**40**) with an unprecedented 14-membered cyclic ether ring. This compound exhibited significant growth inhibitory activity against HepG2 cells with an IC_50_ value of 14.9 µM, highlighting its anticancer potential. A novel anthraquinone derivative, auxarthrols D (**41**), which also exhibited cytotoxic activity against human hepatocellular carcinoma cells with an IC_50_ value of 16.6 µM, was also isolated from the marine fungus *Sporendonema casei* [111]. 

In addition, the actinomycete *Micromonospora matsumotoense* M-412, purified from Aviles Canyon in the Cantabian Sea, produced a novel natural product, paulomycin G (**42**), which was highly cytotoxic to HepG2 cells with an IC_50_ value of 4.30 µg/mL [112]. Dendryphiellin J (**43**), a new burkholderian sesquiterpene isolated from the marine-derived fungus *Cochliobolus lunatus* SCSIO41401, is a rare naturally occurring aldoxime analog with cytotoxicity to HepG-2 cells, with an IC_50_ value of 5.9 μM, respectively [113]. 

Triphenyls are a class of aromatic hydrocarbons consisting of a linear 1,4-diaryl-substituted benzene core with a wide range of biological activities. Researchers identified a triphenyl derivative, CHNQD-00824 (**44**), from the marine fungus *Aspergillus candidus*, which was evaluated for cytotoxic activity against the HepG2 cell line and showed an IC_50_ value of 7.64 µM [114]. The marine-derived fungus *Aspergillus candidus* HM5-4, which was isolated from sponges in the South China Sea, was cultivated on solid rice medium, leading to the production of p-terphenyl derivatives—4″-deoxyterprenin (**45**). Notably, it exhibited cytotoxic effects against human liver cancer cells (BEL-7402), achieving an IC_50_ value of 6.69 μM, thereby demonstrating its potential as a promising anti-cancer agent [115].

The fungal strain BC17 isolated from intertidal sediments in the Gulf of Cadiz was identified as *Emericellopsis maritima*. On the basis of the one strain–many compounds (OSMAC) approach, PR toxin (**46**) was isolated from this strain and it exhibited cytotoxic activity against HepG2 with an IC_50_ value of 8.28 µM [116], further expanding the range of bioactive marine compounds. A strain of the marine fungus *Penicillium* sp. KFD 28 was isolated from the clam Meretrix lusoria from Haikou Bay, and the researchers investigated the secondary metabolites of the fungus and found a number of biologically active indole alkaloids, among which epipaxilline (**47**) showed cytotoxic activity against the human hepatocellular carcinoma cell line BEL-7402. showed cytotoxic activity with an IC_50_ value of 5.3 µM [117]. In addition, a novel compound, monarubin B (**48**), was isolated from the marine shellfish fungus *Monascus ruber* BB5. This compound showed strong cytotoxicity against the HepG2 cancer cell line with an IC_50_ value of 1.72 µM, which not only adds to the growing library of potential anticancer agents, but also emphasizes the rich biodiversity of marine organisms and their associated fungi as a source of novel bioactive compounds [118]. A new thioketopiperazine alkaloid, 7′-demethoxyrostratin C (**49**), was isolated and characterized from culture extracts of the fungus *Epicoccum nigrum* SD-388 obtained from deep-sea sediments (−4500 m). It showed strong cytotoxic activity against Huh7.5 hepatocellular carcinoma cells, and the disulfide bridge at C-2/C-2′ may be the key to its activity, which provides a useful candidate for further investigation of antitumor drugs [119].

Moreover, treatment with seaweed extract triggers a p53-mediated response at the transcriptional and protein levels in liver cancer cells, and this response is linked to the inhibition of cellular proliferation and induction of cell death [120]. *Tisochrysis lutea*, a marine haptophyte, is a veritable repository of omega-3 polyunsaturated fatty acids, such as the essential DHA, and carotenoids, including fucoxanthin. Recent studies have demonstrated that crude extracts of this marine microalga have the potential to serve as a source of bioactive compounds for the prevention and potential therapeutic management of human hepatocarcinoma. Specifically, researchers isolated two cytotoxic and highly selective compounds, loliolide (**50**) and epi-loliolide (**51**), which demonstrated significant efficacy in reducing the viability of HepG2 cancer cells. Notably, these compounds showed significantly reduced toxicity to mouse non-carcinogenic stromal cells, thus highlighting their potential as selective therapeutic agents for hepatocellular carcinoma [121]. In addition, saringosterol acetate (SSA) (**52**), another compound isolated from the edible brown alga *Hizikia fusiforme*, has shown significant anti-tumor effects. SSA could effectively inhibit the growth and metastasis of HCC by targeting cytokines such as IL-6, TNF-*α* and TGF-*β* as well as MMPs. This inhibition is achieved by regulating signaling pathways such as the PI3K/AKT/mTOR and TGF*β*/Smad pathways [122]. The chemical structures of compounds **33**–**52** are shown in Figure 8. These marine extracts and the compounds isolated from them provide new therapeutic strategies for the treatment of hepatocellular carcinoma with small IC_50_ or EC_50_ values (Table 3), and future research in this field is expected to yield more promising anticancer discoveries.

## 3. Conclusions and Future Perspectives

The effect of marine natural products on liver disease is a complex and extensive area of research, and natural products such as alkaloids, PBDEs, polysaccharides, ketones, phenols, and steroids have all been shown to have a positive effect on liver disease. Alkaloids are a class of nitrogen-containing organic compounds widely found in nature, and they usually have complex ring structures, such as indole, piperazine, and pyrazine. These unique structures endow alkaloids with antiviral activity and show great potential in anti-HCV drug development. PBDEs are a class of compounds with multifunctional properties; they not only possess antibacterial, antifungal and anti-inflammatory activities, but also inhibit the enzymatic activities of endogenous and viral proteins. These properties make PBDEs valuable for the development of anti-HBV drugs. Sulfated polysaccharide-like components of marine algae have a variety of biological activities, including antioxidant, anti-inflammatory and metabolic regulation effects, due to their uniqueness, which is of great scientific significance in the treatment of alcoholic fatty liver disease, and provides strong support for the development of novel therapeutic methods and functional foods.

Based on their unique chemical structures, ketone components have important functions in regulating lipid metabolism, especially by acting on metabolic regulators such as LXR*α* and PPAR*α*, which can effectively improve hepatic fat accumulation and inflammatory responses. These properties have enabled ketone components to show significant therapeutic potential in the treatment of nonalcoholic fatty liver disease, providing new ideas and methods for the prevention and treatment of related diseases.

The carotenoids in astaxanthin have shown powerful effects in the treatment of liver fibrosis, acting on various aspects of fibrosis through different mechanisms, providing new strategies and choices for the treatment of liver fibrosis, which is promising in the treatment of liver fibrosis. In addition, carotenoids have been found to be natural products for the treatment of NAFLD, and in general, carotenoids are natural products worthy of further research. In the treatment of liver-injury diseases, phenolic derivatives are able to protect hepatocytes from damage by inhibiting oxidative stress and inflammatory responses, indicating promising applications.

Steroids have unique branched chain structures, such as cyclopropane rings or shorter side chains, and are capable of exerting anti-inflammatory and immunomodulatory effects in the treatment of hepatocellular carcinoma. In addition, triptych derivatives have a wide range of biological activities, including cytotoxicity, *α*-glucosidase inhibitory activity, and AMPK activator activity. Terpenes show great potential in drug discovery due to their unique structural features such as complex tetracyclic structures, sesquiterpene quinone skeletons and acyclic sesquiterpene skeletons. These structural features not only enrich the diversity of terpenoids, but also provide potential new drug candidates for the treatment of hepatitis, liver cancer and other diseases. These bioactivities make these triptych derivatives potentially useful for the development of anti-hepatocellular carcinoma drugs. There may be differences in the types and mechanisms of action of different compounds, so it is important to study their biological activities according to their combination with different targets, so that we can develop safer and more effective drugs for the treatment of liver disease, and so that we can better utilize the role that marine natural product compounds will play in the field of liver disease treatment.

Although marine natural products have made significant progress in the field of liver disease treatment, they still face many challenges. Future research needs to deeply explore their pharmacological mechanisms, optimize the synthesis process, and improve drug purity and activity. Meanwhile, modern biotechnological methods such as gene editing and genomics should be utilized to further study the interaction between marine natural products and liver disease cells, discover new therapeutic targets, and improve the therapeutic effect by combining with traditional therapeutic methods. Validating the safety and effectiveness of the drugs through clinical trials will lay the foundation for the clinical application of marine natural products in the field of liver disease treatment, which will make a greater contribution to human health. We are looking forward to the discovery and application of more effective marine natural products in the treatment of liver diseases with the advancement of science and technology and the deepening of research. 

## Figures and Tables

**Figure 1 marinedrugs-22-00288-f001:**
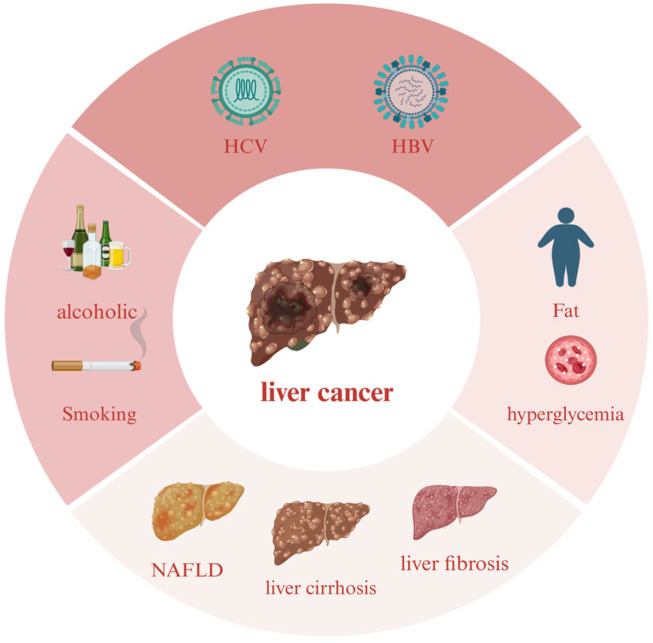
The main factors leading to liver cancer. HBV and HCV are the most important causative factors for liver cancer infection, followed by poor lifestyle habits such as long-term alcohol consumption and metabolic diseases such as obesity, while NAFLD, cirrhosis, and liver fibrosis can also cause liver cancer through lesions.

**Figure 2 marinedrugs-22-00288-f002:**
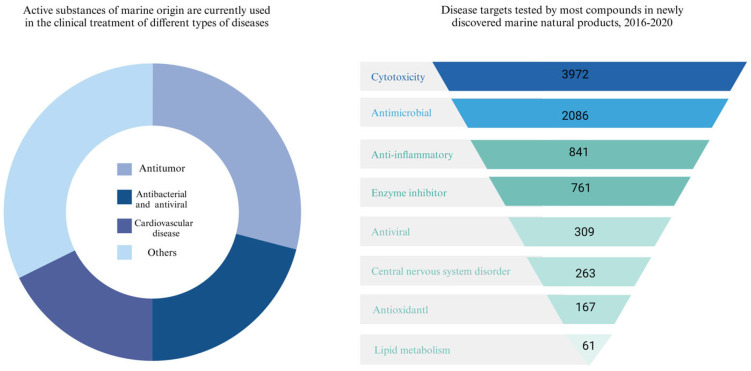
Types of marine drugs in clinical use and current status of research.

**Figure 3 marinedrugs-22-00288-f003:**
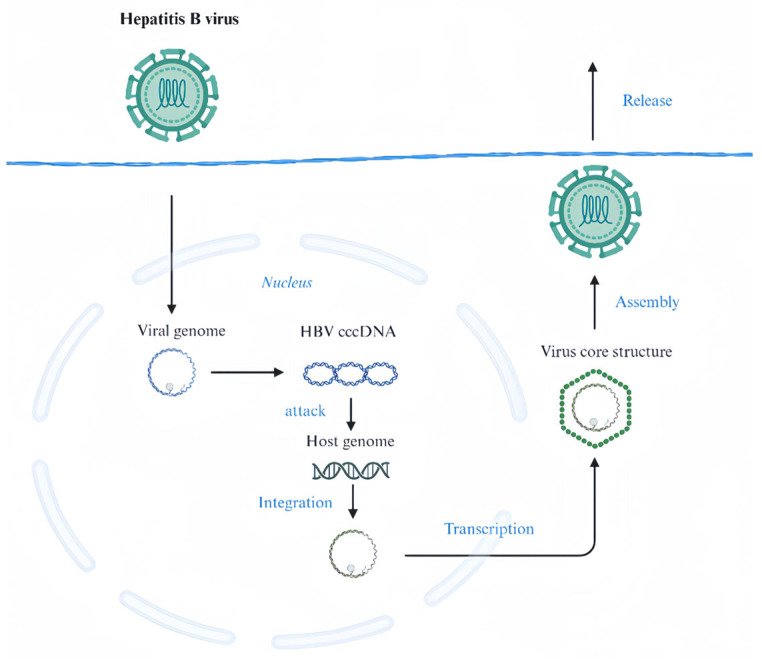
Mechanism of HBV infection.

**Figure 4 marinedrugs-22-00288-f004:**
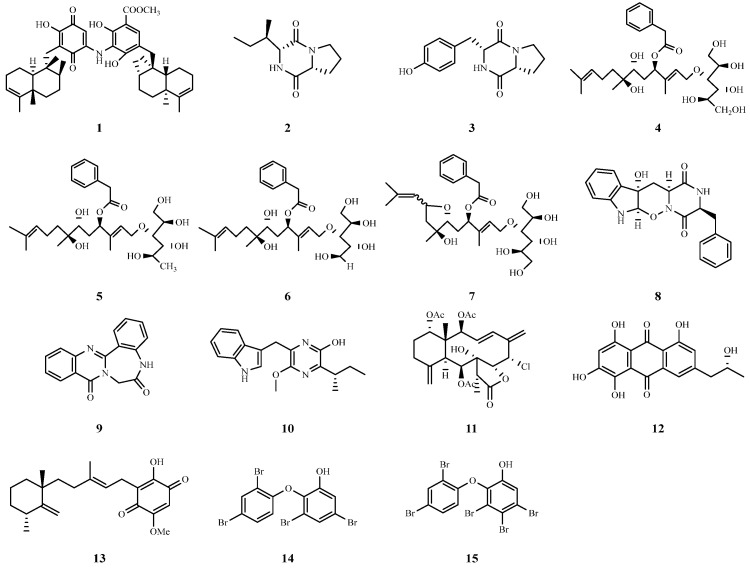
Chemical structures of compounds **1**–**15**.

**Figure 5 marinedrugs-22-00288-f005:**
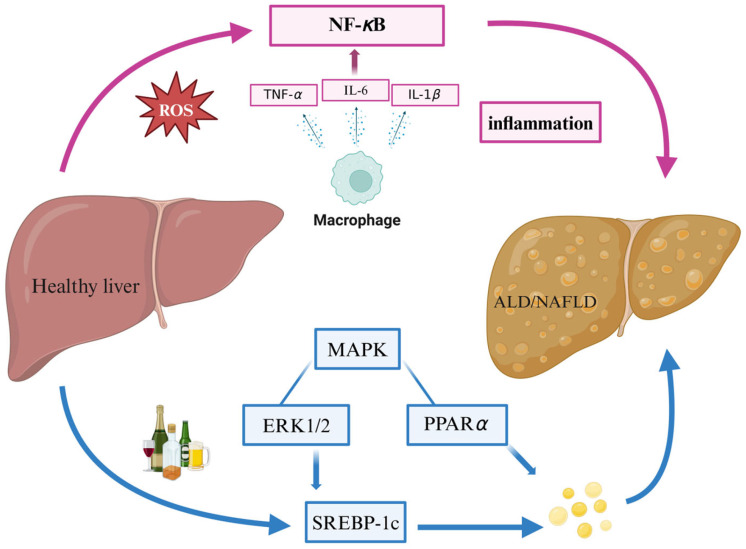
Mechanism of action of marine natural products on ALD and NAFLD.

**Figure 6 marinedrugs-22-00288-f006:**
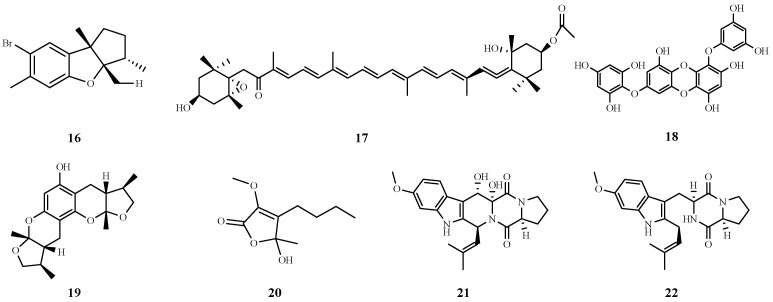
Chemical structures of compounds **16**–**22**.

**Figure 7 marinedrugs-22-00288-f007:**
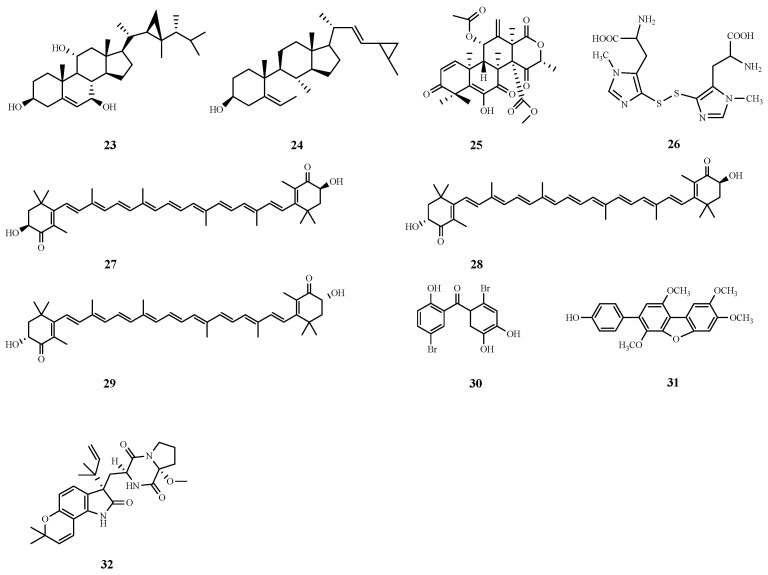
Chemical structures of compounds **23**–**32**.

**Figure 8 marinedrugs-22-00288-f008:**
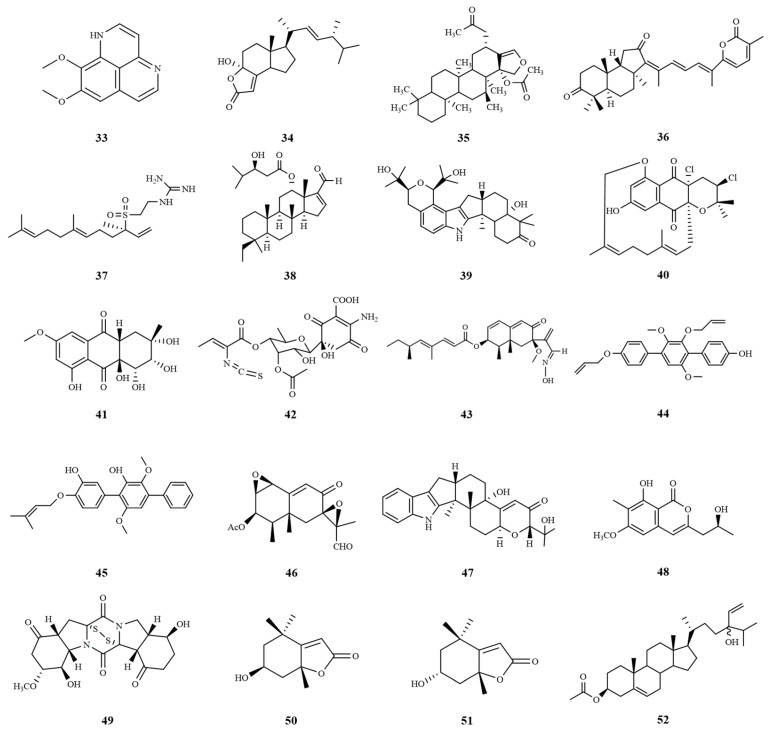
Chemical structures of compounds **33**–**52**.

**Table 1 marinedrugs-22-00288-t001:** IC_50_ or EC_50_ of compounds on HCV and HBV.

Number	Name	Source	Disease	IC_50_ (μM)	EC_50_ (μM)
**1**	Nakijiquinone F	*Spongia irregularis*	HCV	12.6	—
**2**	Cyclic (L-tyrosine-L-proline)	*Aspergillus versicolor*	HCV	8.2	—
**3**	Cyclic (L-proline-L-enine)	*Aspergillus versicolor*	HCV	13.4	—
**4**–**7**	Acremosides A and C–E	*Acremonium* sp. IMB18-086	HCV	—	4.8~8.8
**8**	Raistrickindole A	*Penicillium raistrickii* IMB17-034	HCV	—	5.7
**9**	Raistrickin	*Penicillium raistrickii* IMB17-034	HCV	—	7.0
**10**	Trypilepyrazinol	*Penicillium* sp. IMB 17-046	HCV	7.7	—
**12**	(−)-2′*R*-1-hydroxyisorhodoptilometrin	*Penicillium* sp. OUCMDZ-4736	HBV	4.63	—
**13**	Metachromin A	*Dactylospongia metachromia*	HBV	—	0.8
**14**	3,5-dibromo-2-(2,4-dibromophenoxy)-phenol	*Dysidea* sp.	HBV	—	0.23
**15**	3,4,5-tribromo-2-(2,4-dibromophenoxy)-phenol	*Dysidea* sp.	HBV	—	0.80

**Table 2 marinedrugs-22-00288-t002:** Effects of compounds on ALD, NAFLD, liver fibrosis and liver injury.

Number	Name	Source	Disease	Effects
**16**	Aplysin	*Laurencia tristicha*	ALD	Anti-oxidation Bcl-2, Bax
**17**	Fucoxanthin	*Laminaria japonica* Aresch	ALD	AST, ALT, KEAP1/Nrf2/ARE PGC1*α*/NRF1
**18**	7-Phloroeckol	*Ecklonia cava*	ALD	GSH, SOD, Bcl-2
**19**	Xyloketal B	*Xylaria* sp.	NAFLD	PPAR*α*/PGC1*α*
**20**	Furazone	*Setosphaeria* sp. SCSIO41009	NAFLD	LXR*α*, PPAR*α*
**21**	12R,13S-dihydroxyfumitremorgin C	*Aspergillus* sp. SCSIO 41420	NAFLD	LXR*α*
**22**	Tryprostatin A	*Aspergillus* sp. SCSIO 41420	NAFLD	LXR*α*
**23**	Dissesterol	*Sarcophyton glaucum*	Liver fibrosis	GST, SOD
**24**	Glaucasterol	*Sarcophyton glaucum*	Liver fibrosis	GST, SOD
**25**	Aspermeroterpene B	*Aspergillus terreus* GZU-31-1	Liver fibrosis	Nrf2
**27**	3S-3′S Astaxanthin	*Haematococcus* *pluvialis*	Liver fibrosis	Anti-oxidation NrF2 TGF-*β*1/Smad3
**28**	3R-3′S Astaxanthin			
**29**	3R-3′R Astaxanthin			
**30**	LM49	*Galaxaura oblongata*	Liver injury	NF-*κ*B, MPO, LPO
**31**	HN-001	*Aspergillus* sp. C1	Liver injury	PLA2/IRE-1*α*/XBP-1s JNK
**32**	Notoamide Q	*Aspergillus amoenus* TJ507	Liver injury	ALT, AST, LDH

**Table 3 marinedrugs-22-00288-t003:** IC_50_ values of the compounds against HCC.

Number	Name	Source	IC_50_ (μM)
**38**	Phyllospongiane C	*Phyllospongia foliascens*	9.6
**40**	Near-tearomycin D1	*Streptomyces* sp. strain CA-271078	14.9
**41**	Auxarthrols D	*Sporendonema casei*	16.6
**42**	Paulomycin G	*Micromonospora matsumotoense* M-412	4.3
**43**	Dendryphiellin J	*Cochliobolus* lunatus SCSIO41401	5.9
**44**	CHNQD-00824	*Aspergillus candidus* (CHNSCLM-0393)	7.64
**45**	4″-Deoxyterprenin	*Aspergillus candidus* HM5-4	6.69
**46**	PR toxin	*Emericellopsis maritima* BC17	8.28
**47**	Epipaxilline	*Penicillium* sp. KFD 28	5.3
**48**	Monarubin B	*Monascus ruber* BB5	1.72

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
