# Peer review of "Effects of Marine Natural Products on Liver Diseases"

_marinedrugs, 2024, doi:10.3390/md22070288_

Round 1
Reviewer 1 Report
Comments and Suggestions for Authors
1. Figure 1 only lists the factors that cause liver cancer. It is suggested to rank or label the top three factors that cause liver cancer and add necessary discussion.
2. What is the special meaning of the colored markings on the chemical structures of compounds 9-21 in Figure 2? If no, it should be deleted it. Other color labels for alkaloids, polyketones and phenols are also recommended to be removed.
3. Most of chemical structures in Figure 2 are not clearly drawn, especially the chemical bonds indicating the configuration. Compound 20 with the wrong structure belongs to isocoumarin, and compound 35 with the wrong structure belongs to indole diterpene. Please check all the chemical structures in Figure 2 carefully for errors.
4. In the title of Figure 2, 1-23 are new compounds, but they are not classified. Compounds 54 and 17-18 are of the same type and should not be drawn separately. Bromine compounds are suggested to be drawn together. Therefore, the chemical structures in Figure 2 should be rearranged according to the structure types.
5. Table 2 and Table 3 are not related to the chemical structures in Figure 2, so it is suggested to add the serial numbers of the compounds to the tables.
6. What are the common structural characteristics of the active compounds? A brief summary discussion is recommended.
Comments on the Quality of English Language
1. Line 105,There is an error message.
2. References are cited in inconsistent formats in the text.
3. The serial numbers of the compounds should be shown in bold.
4. The names of compounds in Tables 2 and 3 should start with uppercase letters, and the Latin names of species should be italicized.
Author Response
Dear Reviewer,
We gratefully thank the editor and reviewer who took the time to make constructive comments and useful suggestions, which greatly improved the quality of the manuscript and enabled us to improve the manuscript. On behalf of all the contributing authors, I would like to express my sincere gratitude for your letter and the constructive comments made by the reviewers on our article entitled "Effects of marine natural products on liver diseases" (Marinedrugs-3058778). Every revision suggestion and comment made by the reviewer was accurately incorporated and considered. Below the reviewer's comments were point-by-point responses. What’s more, we have also reviewed and polished the language of the manuscript throughout, and these changes will not affect the content and framework of the paper. In addition, we have marked the revised paper in red. thanks for your suggestions. I am so sorry to bring you so much trouble because of our careless. Thanks very much again for your attention to our paper. Once again, thank you for your help to our paper processing.
Comments and Suggestions
Question 1:
Figure 1 only lists the factors that cause liver cancer. It is suggested to rank or label the top three factors that cause liver cancer and add necessary discussion.
Answer: Thank you for your valuable suggestions. Considering the Reviewer’s suggestion, we have revised Figure 1 and added a discussion of the key factors that contribute to hepatocellular carcinoma, HBV and HCV are the most important causative factors for liver cancer infection, followed by poor lifestyle habits such as long-term alcohol consumption and metabolic diseases such as obesity, while NAFLD, cirrhosis, liver fibrosis can also form liver cancer through lesions. as detailed in Figure 1 and lines 29-41 of the article.
Question 2:
What is the special meaning of the colored markings on the chemical structures of compounds 9-21 in Figure 2? If no, it should be deleted it. Other color labels for alkaloids, polyketones and phenols are also recommended to be removed.
Answer: Thank you for your suggestion. Thank you for your suggestion. The colored markings on the chemical structures were originally intended only to reflect their common structural features. After discussion, we have deleted the original Fig. 2 and replaced it with Fig. 2, Fig. 4, Fig. 6, and Fig. 7, and eliminated the colored markings from the chemical structures. See Figures 2, 4, 6 and 7 for details.
Question 3: Most of chemical structures in Figure 2 are not clearly drawn, especially the chemical bonds indicating the configuration. Compound 20 with the wrong structure belongs to isocoumarin, and compound 35 with the wrong structure belongs to indole diterpene. Please check all the chemical structures in Figure 2 carefully for errors.
Answer: We apologize for our oversight in the chemical structure. We have redrawn the chemical structure and checked all the structures in the manuscript. See Figures 2, 4, 6 and 7 for details.
Question 4: In the title of Figure 2, 1-23 are new compounds, but they are not classified. Compounds 54 and 17-18 are of the same type and should not be drawn separately. Bromine compounds are suggested to be drawn together. Therefore, the chemical structures in Figure 2 should be rearranged according to the structure types.
Answer: Thank you for your nice comments on our article. Due to our improper arrangement of the compounds, after discussion, we have deleted Fig. 2 and drawn the corresponding compounds of the segment under each type of disease, i.e. Fig. 2, Fig. 4, Fig. 6 and Fig. 7, and summarized and discussed the types of compounds and treatment of the diseases at the conclusion of the article, which is shown in lines 708-742 of the article.
Question 5: Table 2 and Table 3 are not related to the chemical structures in Figure 2, so it is suggested to add the serial numbers of the compounds to the tables.
Answer: We sincerely appreciate your valuable comments and we have added the serial numbers of the compounds to the table as detailed in Tables 2 and 3.
Question 6: What are the common structural characteristics of the active compounds? A brief summary discussion is recommended.
Answer: Thank you for pointing this out. The reviewer is correct, and we have added a discussion of the structural characterization of the active compounds at the end, as detailed in lines 708-742 of the article.
Comments on the Quality of English Language
Question 1: Line 105,There is an error message.
Answer: We apologize profusely for the misspelling and have corrected it.
Question 2: References are cited in inconsistent formats in the text.
Answer: We apologize for our negligence in reference formatting and we have carefully checked and standardized our reference formatting.
Question 3: The serial numbers of the compounds should be shown in bold.
Answer: We were really sorry for our careless mistakes. Thank you for your reminder. We have bolded the serial numbers of all compounds.
Question 4: The names of compounds in Tables 2 and 3 should start with uppercase letters, and the Latin names of species should be italicized.
Answer: We feel sorry for our carelessness. We have changed the compound names in Tables 2 and 3 to begin with a capital letter and the Latin names of species to italics, and checked the compound names and Latin names of species throughout the text.
Thank you again for your positive comments and valuable suggestions to improve the quality of our manuscript.
We submit here the revised manuscript as well as a list of changes.
If you have any question about this paper, please don’t hesitate to let me know. Thank you very much for your attention and consideration to our revised manuscript.
Best wishes.
Sincerely yours.
Haifeng Wang, Ph. D.
Professor
Shenyang Pharmaceutical University
Wenhua Road 103
Shenyang 110016, P R. China
Tel: +86-24-23980791
Fax: +86-24-23980791
Email: wanghaifeng0310@163.com

Reviewer 2 Report
Comments and Suggestions for Authors
The topic of this manuscript is quite interesting. When I received this manuscript, I immediately set aside time to read this manuscript. This is a comprehensive review focus on the different groups of marine natural products from a vast array of marine natural products, displaying various types of pharmacological effects against liver diseases. There is no doubt that this manuscript is worth publishing in this journal.
However, revisions were required as followings:
1. What is the time span for this review? It is important information for readers.
2. The bioassays for marine natural products in the last five years include not only cytotoxicity, antimicrobial, anticancer, antiviral, and lipid metabolism, but also neuroprotective, antifouling, antioxidant, and others. The right half of Table 1 should be modified. Additionally, the Table 1 was more likely a figure not a table.
3. What did you mean ‘novel structures? Usually, a novel structure possesses a new or extremely rare carbon skeleton. However, compounds 1–23 were not. The classifications of compounds in Figure 1 should be revised.
4. There were lack of information for some compounds. For example, what were the novel compounds from the marine sponge Amphimedon sp.?(P5L116), what were the two secondary metabolites from the fungus Fusarium equiseti?(P5L144)
5. The numbers of these compounds should be reorganized in accordance with the order appeared in the manuscript. The first mentioned compound should be numbered 1, followed by sequential numberings for other compounds. However, these numbers were in a mess. For instance, the number for nakijiquinone F was 36, but it was mentioned before cyclic (L-proline-L-enine), whose number was 24. Moreover, these numbers should be in bold. Consequently, the organizations of compounds shown in Figure 1 should be adjusted.
6. What was the purpose of providing Table 2 in the manuscript? Not all the structures described in the subsection 2.1 were listed in this table.
7. The source organisms of these secondary metabolites should be clearly reported in this manuscript. For instance, the source organism for compounds 3,5-dibromo-2-(2,4-dibromophenoxy)-phenol and 3,4,5-tribromo-2-(2,4-dibromophenoxy)-phenol was the marine sponge Dysidea sp., which was clearly pointed out in the Ref. [37]. Indeed, Indonesian coral reefs consist of a vast array of organisms including corals, sponges, mollusks, and others. Please check this type of information for other parts of this manuscript. For instance, the fungal strain BC17 should be clearly assigned as Emericellopsis maritima BC17 in Table 3.
8. Why there was no tables showing the summary of compounds with specific activities in the following subsections 2.2–2.5?
9. Since these reported in this manuscript had been clarified into different groups of compounds, it is better to give a summary of pharmacological effects against liver diseases in the Conclusions.
Others:
1. The list of authors need to be revised. And there was no author of the second affiliation
2. P5L105: Error for the reference citation
3. Italic font for the species name, such as ‘Penicillium raistrickii’(P5L136) and ‘Cochliobolus lunatus’(P16L622)
4. There were many references lack of pages at the end of this manuscript, such as [57], [66], [71], [90], [113], [116].
Comments on the Quality of English Language
There were a few typo and grammar errors, some of which were given in the comments.
Author Response
Dear Reviewer,
We gratefully thank the editor and reviewer who took the time to make constructive comments and useful suggestions, which greatly improved the quality of the manuscript and enabled us to improve the manuscript. On behalf of all the contributing authors, I would like to express my sincere gratitude for your letter and the constructive comments made by the reviewers on our article entitled "Effects of marine natural products on liver diseases" (Marinedrugs-3058778). Every revision suggestion and comment made by the reviewer was accurately incorporated and considered. Below the reviewer's comments were point-by-point responses. What’s more, we have also reviewed and polished the language of the manuscript throughout, and these changes will not affect the content and framework of the paper. In addition, we have marked the revised paper in red. thanks for your suggestions. I am so sorry to bring you so much trouble because of our careless. Thanks very much again for your attention to our paper. Once again, thank you for your help to our paper processing.
Reviewer
The topic of this manuscript is quite interesting. When I received this manuscript, I immediately set aside time to read this manuscript. This is a comprehensive review focus on the different groups of marine natural products from a vast array of marine natural products, displaying various types of pharmacological effects against liver diseases. There is no doubt that this manuscript is worth publishing in this journal.
Answer: First of all, thank you for your recognition of our manuscript. We greatly appreciate your constructive comments and useful suggestions, which greatly improves the quality of the manuscript. Each suggested revision and comments, brought forward by you was accurately incorporated and considered. All changes in the manuscript are marked using red font.
Comments and Suggestions
Question 1:
What is the time span for this review? It is important information for readers.
Answer: We apologize for our oversight of the time span of this review, which we have added to the abstract section, as detailed in line 19 of the article.
Question 2:
The bioassays for marine natural products in the last five years include not only cytotoxicity, antimicrobial, anticancer, antiviral, and lipid metabolism, but also neuroprotective, antifouling, antioxidant, and others. The right half of Table 1 should be modified. Additionally, the Table 1 was more likely a figure not a table.
Answer: We sincerely appreciate the valuable comments. After reviewing the literature, we have added figures to Table 1 and made additions to Table 1.
Question 3: What did you mean ‘novel structures? Usually, a novel structure possesses a new or extremely rare carbon skeleton. However, compounds 1–23 were not. The classifications of compounds in Figure 1 should be revised.
Answer: We have rewritten this section based on the reviewer's suggestion to remove Figure 1 and reclassify the compounds as detailed in lines 708-742 of the article.
Question 4: There were lack of information for some compounds. For example, what were the novel compounds from the marine sponge Amphimedon sp.? (P5L116), what were the two secondary metabolites from the fungus Fusarium equiseti?(P5L144)
Answer: We apologize for the lack of compound information, after reviewing the literature, the ethyl acetate extract from the marine sponge Amphimedon sp. did not address the structure of the specific new compounds, so we decided to delete this section, and the two secondary metabolites of Fusarium xylophilum did not match the time span of the current review, so we also deleted this section. In addition, we checked the completeness of the information on the other compounds in this paper.
Question 5: The numbers of these compounds should be reorganized in accordance with the order appeared in the manuscript. The first mentioned compound should be numbered 1, followed by sequential numberings for other compounds. However, these numbers were in a mess. For instance, the number for nakijiquinone F was 36, but it was mentioned before cyclic (L-proline-L-enine), whose number was 24. Moreover, these numbers should be in bold. Consequently, the organizations of compounds shown in Figure 1 should be adjusted.
Answer: Apologies for the inappropriate order in which the compound numbers appeared, we have reorganized them and they are shown in bold, replacing the original Figure 1 with Figures 2, 4, 6 and 7.
Question 6: What was the purpose of providing Table 2 in the manuscript? Not all the structures described in the subsection 2.1 were listed in this table.
Answer: Thank you for pointing this out. Table 2 summarizes the IC50 or EC50 values of many compounds against HCV and HBV, which provide a more intuitive representation of the antiviral capacity of these compounds through numerical values, while other compounds did not have a specific IC50 or EC50 determined, but were represented by other tests.
Question 7: The source organisms of these secondary metabolites should be clearly reported in this manuscript. For instance, the source organism for compounds 3,5-dibromo-2-(2,4-dibromophenoxy)-phenol and 3,4,5-tribromo-2-(2,4-dibromophenoxy)-phenol was the marine sponge Dysidea sp., which was clearly pointed out in the Ref. [37]. Indeed, Indonesian coral reefs consist of a vast array of organisms including corals, sponges, mollusks, and others. Please check this type of information for other parts of this manuscript. For instance, the fungal strain BC17 should be clearly assigned as Emericellopsis maritima BC17 in Table 3.
Answer: We apologize for our oversight on the biological sources of secondary metabolites. We have revised the text as well as Tables 2 and 3 for the biological sources of the compounds you mentioned, and we have checked the sources of the other secondary metabolites in this paper.
Question 8: Why there was no tables showing the summary of compounds with specific activities in the following subsections 2.2–2.5?
Answer: Thank you for your valuable comments on our article. For subsections 2.2-2.5, we have added the effects of compounds on ALD, NAFLD, Liver fibrosis and Liver injury. see Table 3 for details.
Question 9: Since these reported in this manuscript had been clarified into different groups of compounds, it is better to give a summary of pharmacological effects against liver diseases in the Conclusions.
Answer: We feel great thanks for your professional review work on our article. As you are concerned, we have added a summary of the pharmacological effects of different types of compounds on liver disease to our conclusions. See lines 708-742 of the article for details.
Others:
Question 1: The list of authors need to be revised. And there was no author of the second affiliation.
Answer: Thank you for pointing this out. We have revised the list of authors. See the third line of the article for details.
Question 2: P5L105: Error for the reference citation
Answer: We apologize profusely for the misspelling and have made corrections.
Question 3: Italic font for the species name, such as ‘Penicillium raistrickii’(P5L136) and ‘Cochliobolus lunatus’(P16L622)
Answer: We apologize for our neglect of the italicization of species names, which you mention has been corrected, as detailed in lines 639 and 143, and in addition, we have checked the italicization of species names throughout the text.
Question 4: There were many references lack of pages at the end of this manuscript, such as [57], [66], [71], [90], [113], [116].
Answer: We feel sorry for our carelessness. We have completed the page numbers of the references you mentioned and checked all references.
Comments on the Quality of English Language
Question: There were a few typo and grammar errors, some of which were given in the comments.
Answer: We apologize for the typos and grammatical errors and have checked and corrected them throughout the text.
Thank you again for your positive comments and valuable suggestions to improve the quality of our manuscript.
We submit here the revised manuscript as well as a list of changes.
If you have any question about this paper, please don’t hesitate to let me know. Thank you very much for your attention and consideration to our revised manuscript.
Best wishes.
Sincerely yours.
Haifeng Wang, Ph. D.
Professor
Shenyang Pharmaceutical University
Wenhua Road 103
Shenyang 110016, P R. China
Tel: +86-24-23980791
Fax: +86-24-23980791
Email: wanghaifeng0310@163.com

Round 2
Reviewer 1 Report
Comments and Suggestions for Authors
The manuscript was properly revised according to the reviewer's comments, but all chemical structures are suggested to be applied setting from ACS document 1996. This manuscript is acceptable to be published in Marine Drugs.
Comments on the Quality of English Language
It's OK.
Reviewer 2 Report
Comments and Suggestions for Authors
This manuscript has been improved by authors. However, revisions were still required.
1. In the manuscript, it was said ‘Xylitol B (Xyl-B) (19) is a natural marine ketone’. However, there was no carbonyl group in the structure. According, Xyl-B cannot be classified as a ketone. Moreover, authors drew the chemical structure of Xyl-B wrongly. Compared with the structure shown in the reference [61], one oxygen atom was missing. Please check all the chemical structures and their classifications throughout the whole manuscript.
2. There were many terpenes reported, including the meroterpene nakijiquinone F (1), the sesquiterpene acremoside A (4), the sesterterpene phyllospongiane C (38). However, there was no discussion on terpenes in the Conclusions.
Others:
1. No need to capitalize the first letter of compound names unless it is at the beginning of a sentence. For example, revise ‘among which Nakijiquinone F (1) showed…’ as ‘among which nakijiquinone F (1) showed…’.
2. Figures 2, 4, 6, and 7 captions: ‘Chemical structure of compounds’ → ‘Chemical structures of compounds’
3. Tables 2 & 4: Please list compounds in ascending order.
4. P4L138: ‘Acremosides A (4) and Acremosides C-E (5-7)’ → ‘acremoside A (4) and C–E (5–7)’
5. Please check the references more carefully. For example, the source journals of [61] and [63] were identical, but one used its full name, the other used its abbreviation.
Comments on the Quality of English Language
There were still a few typo and grammar errors.
Author Response
Dear Reviewer,
On behalf of all the contributing authors, I would like to express my sincere gratitude for your letter and the constructive comments made by the reviewers on our article entitled "Effects of marine natural products on liver diseases" (Marinedrugs-3058778). Every revision suggestion and comment made by the reviewer was accurately incorporated and considered. Below the reviewer's comments were point-by-point responses. In addition, we have marked the revised paper in red. thanks for your suggestions. I am so sorry to bring you so much trouble because of our careless. Thanks very much again for your attention to our paper. Once again, thank you for your help to our paper processing.
Comments and Suggestions
Question 1:
In the manuscript, it was said ‘Xylitol B (Xyl-B) (19) is a natural marine ketone’. However, there was no carbonyl group in the structure. According, Xyl-B cannot be classified as a ketone. Moreover, authors drew the chemical structure of Xyl-B wrongly. Compared with the structure shown in the reference [61], one oxygen atom was missing. Please check all the chemical structures and their classifications throughout the whole manuscript.
Answer: We sincerely appreciate the valuable comments. By reviewing the literature, we have revised the description of Xyl-B. In addition, we apologize for the misdrawing of the chemical structure, which has been redrawn, and we have checked all chemical structures and their classifications throughout the manuscript.
Question 2:
There were many terpenes reported, including the meroterpene nakijiquinone F (1), the sesquiterpene acremoside A (4), the sesterterpene phyllospongiane C (38). However, there was no discussion on terpenes in the Conclusions.
Answer: Thank you for pointing this out. We apologize for the oversight in the terpene summary and have added this section at the end, as detailed in lines 754-758 of the article.
Others:
Question 1: No need to capitalize the first letter of compound names unless it is at the beginning of a sentence. For example, revise ‘among which Nakijiquinone F (1) showed…’ as ‘among which nakijiquinone F (1) showed…’.
Answer: Thank you for pointing this out. We have changed all capital letters to lower case in similar situations, thank you for pointing this out.
Question 2: Figures 2, 4, 6, and 7 captions: ‘Chemical structure of compounds’ → ‘Chemical structures of compounds’
Answer: We apologize profusely for the misspelling and have made corrections.
Question 3: Tables 2 & 4: Please list compounds in ascending order.
Answer: We apologize that the order of compounds in the chart was not in ascending order and have adjusted it to ascending order.
Question 4: P4L138: ‘Acremosides A (4) and Acremosides C-E (5-7)’ → ‘acremoside A (4) and C–E (5–7)’
Answer: We feel sorry for our carelessness. We have adjusted the compound names.
Question 5: Please check the references more carefully. For example, the source journals of [61] and [63] were identical, but one used its full name, the other used its abbreviation.
Answer: We apologize for our carelessness. We have carefully checked all references and made corrections.
Comments on the Quality of English Language
Question: There were still a few typo and grammar errors.
Answer: We apologize for the typos and grammatical errors in the text, which we checked for and corrected throughout.
Thank you again for your positive comments and valuable suggestions to improve the quality of our manuscript.
We submit here the revised manuscript as well as a list of changes.
If you have any question about this paper, please don’t hesitate to let me know. Thank you very much for your attention and consideration to our revised manuscript.
Best wishes.
Sincerely yours.
Haifeng Wang, Ph. D.
Professor
Shenyang Pharmaceutical University
Wenhua Road 103
Shenyang 110016, P R. China
Tel: +86-24-23980791
Fax: +86-24-23980791
Email: wanghaifeng0310@163.com
